# Diacerein-Loaded Hyaluosomes as a Dual-Function Platform for Osteoarthritis Management via Intra-Articular Injection: In Vitro Characterization and In Vivo Assessment in a Rat Model

**DOI:** 10.3390/pharmaceutics13060765

**Published:** 2021-05-21

**Authors:** Nouran O. Eladawy, Nadia M. Morsi, Rehab N. Shamma

**Affiliations:** Department of Pharmaceutics and Industrial Pharmacy, Faculty of Pharmacy, Cairo University, Cairo 11562, Egypt; nouran.abdelmageed@pharma.cu.edu.eg (N.O.E.); nadia.morsi@pharma.cu.edu.eg (N.M.M.)

**Keywords:** diacerein, hyaluronic acid, osteoarthritis, intra-articular injection

## Abstract

The application of intra-articular injections in osteoarthritis management has gained great attention lately. In this work, novel intra-articular injectable hyaluronic acid gel-core vesicles (hyaluosomes) loaded with diacerein (DCN), a structural modifying osteoarthritis drug, were developed. A full factorial design was employed to study the effect of different formulation parameters on the drug entrapment efficiency, particle size, and zeta potential. Results showed that the prepared optimized DCN- loaded hyaluosomes were able to achieve high entrapment (90.7%) with a small size (310 nm). The morphology of the optimized hyaluosomes was further examined using TEM, and revealed spherical shaped vesicles with hyaluronic acid in the core. Furthermore, the ability of the prepared DCN-loaded hyaluosomes to improve the in vivo inflammatory condition, and deterioration of cartilage in rats (injected with antigen to induce arthritis) following intra-articular injection was assessed, and revealed superior function on preventing cartilage damage, and inflammation. The inflammatory activity assessed by measuring the rat’s plasma TNF-α and IL-1b levels, revealed significant elevation in the untreated group as compared to the treated groups. The obtained results show that the prepared DCN-loaded hyaluosomes would represent a step forward in the design of novel intra articular injection for management of osteoarthritis.

## 1. Introduction

Osteoarthritis (OA) is the most common type of arthritis, and one of the most important degenerative joint disease causing disability among elderly people worldwide [1]. Worldwide, OA affects 303 million people in 2017 [2], with 9.6% of men and 18.0% of women aged over 60 years suffer from OA symptoms [3]. OA can affect any joint, but especially affects the knee, hip, hands, and spine, resulting in pain and disability of the patient, and an economic burden on society.

OA is a complex disease; whose pathogenesis alters homeostasis of the articular cartilage tissue and subchondral bone. In OA, the bone becomes stiffer and its ability to absorb impact loads is decreased, resulting in more stress on the cartilage. Among the common features of OA are cartilage loss, narrowing of joint space, hypertrophic bone changes [4]. The most common symptom of OA is chronic pain. Both disability and pain contributes to a significant reduction in life quality [5]. Unfortunately, most of the pharmacological therapies used in OA management are symptomatic. The first-line oral analgesic used in OA pain management is paracetamol, while oral non-steroidal anti-inflammatory drugs are the keystone of therapy [6]. Although these drugs are recommended by the societies of rheumatology and governmental agencies, they are considered as symptom managing rapid-acting drugs in OA while the treatment of the actual cause of pain remains neglected.

Diacerein (DCN) is considered as a structural modifying OA drug that can specifically hinder interleukins-1b (IL-1b) in human monocytes which has an important effect on the degeneration of cartilage [7]. DCN is commonly used to manage the OA symptoms, target the principal causes, and thus hinder the progress of the disease. However, being a BCS class II drug, its slight solubility in water (3.197 mg/L) and poor oral bioavailability (35–56%) [8] limits its beneficial effect in OA treatment. Moreover, the remaining unabsorbed drug after oral administration results in a laxative effect via stimulation of prostaglandins and release of acetylcholine as well, therefore results in diarrhea [9]. Furthermore, DCN oral treatment requires frequent administration because of its short half-life (4–8 h) [10], together with its undesirable laxative effect can lead to difficulty in patient compliance and adherence to DCN oral regimen. Many studies have been reported on DCN formulation in colloidal systems in order to avoid undesirable oral treatment effects such that reported by Aziz DE et al., [11] in their study on elastosomes for boosting the transdermal delivery of DCN, and that reported by El-Say KM et al., [12] in their study on DCN niosomal gel for topical delivery.

Intra-articular (IA) drug delivery is a new strategy in management of OA. IA drug delivery aims to increase lubrication, reduce pain, and enhance the flexibility in the joint [13]. IA drug delivery uniqueness returns to the joint anatomy itself. Injection of drugs directly into the joint resulted in the release of drugs locally in the target area. Accordingly, decrease the amount of the drug needed to achieve the desired pharmacological activity at the site of action and decrease drug exposure to unwanted sites [14].

Hyaluronic acid (HA), a nonsulfated glycosaminoglycan, is an important component of the extracellular matrix. HA represents a major component of articular cartilage and synovial fluid. HA (also known as hyaluronan) is utilized in treatment of joint diseases, such as OA and rheumatoid arthritis [15], where these disorders characterized by decreasing both the concentration and molecular weight of HA in the synovial fluid, leading to decreasing the elasticity and viscosity of the synovial fluid as well. The affected synovial fluid cannot perform its role in cushioning the knee joint, leading to cartilage wearing down with time. This decay can cause pain and stiffness of the knee [16]. Such treatments, called visco-supplementation treatments, are delivered as IA injections in the knee joint and are considered to restore the viscosity of the joint fluid, resulting into lubricating and cushioning the joint, and can also produce an analgesic effect by balancing the load transmission across the articular surfaces. HA specifically forms stable aggregate via interacting with cartilage proteoglycans, thus helps in suppling nourishment and removal of waste from the articular matrix and prevents excess fibrous tissue from existing in the matrix of cartilage. HA can also add an anti-inflammatory effect to the synovial fluid [17].

Liposomes are defined as vesicles of phospholipids contain one or more concentric lipid bilayers include aqueous core. The uniqueness of liposomes returns to their ability to entrap both lipophilic and hydrophilic compounds, thus allowing encapsulation of a diverse range of drugs by these vesicles. The lipid bilayer can enclose the hydrophobic molecules, while the aqueous core can entrap the hydrophilic molecules [18]. Soybean lecithin has been chosen as natural phospholipid to be used in the preparation of liposomes in many studies, since soya lecithin has many advantages such as, having higher stability (less polyunsaturated fatty acids), safety, availability in both purified and non-purified forms, and allowing cost reduction on both laboratory and pharmaceutical level of production of liposomes [19]. However, in general liposomes as delivery system was susceptible to fast elimination from the bloodstream, which limits its therapeutic activity. Thus, in order to enhance liposome stability and prolong their time of circulation in the blood, liposomes could be sterically stabilized by the addition of some polymers such as HA to the liposomes [20]. Hyaluronic acid gel-filled core liposomes (Hyaluosomes) were previously studied [21] as novel liposomes that are consisted of phospholipid vesicles encapsulating gel material in their core. Hyaluosomes can take the advantages of liposomes and gel formulations in one drug delivery system [21,22,23]. The articular viscoelasticity can be restored via injection of exogenous HA into a joint, having a good response on the cartilage affected due to OA damage, this may be related to the main effect of HA on the cartilage surface as mentioned. Therefore, this study aims to combine the advantages of the OA drug, DCN, being loaded into HA gel- core filled liposomes in OA treatment, through the development of DCN-loaded hyaluosomes to be administered via IA injection. The properties of the formed hyaluosomes will be optimized by varying different formulation variables and studying their effects on the drug entrapment efficiency, particle size, and zeta potential of the prepared DCN-loaded hyaluosomes using a full factorial design. Design expert program will be used to select the optimized system. The morphology of the optimized DCN-loaded hyaluosomes were examined using transmission electron microscopy, and the possibility of any chemical or physical interaction between the drug and the excipients will be studied using FTIR and DSC, respectively. Finally, the optimized system further assessed for their ability to improve the inflammatory condition, and deterioration of cartilage in rats that were injected with antigen to induce arthritis.

## 2. Materials

Diacerin (DCN) was obtained from EVA Pharmaceutical Industries (Cairo, Egypt). Sodium hyaluronate, 95%, (molecular weight 1.5 to 2.2 million Da) was purchased from Acros Organics Co., Geel, Belgium. Cellulose dialysis membrane average width 33 mm (1.3 in), (typical molecular weight cut-off = 14,000 Da), Ovalbumin, Freund’s adjuvant, Soya Lecithin, Hematoxylin and Eosin were purchased from Sigma Aldrich Co., St. Louis, MO, USA. Tween 80%, Sodium Chloride (NaCl), Potassium Chloride (KCl), Disodium hydrogen Phoshate (Na_2_HPO_4_), Potassium Dihydrogen Phosphate (KH_2_PO_4_), Ethanol, and Xylene, were purchased from Adwic, El- Nasr pharmaceutical company, Cairo, Egypt. Cal-Ex II Decalcifier, were purchased from Fisher Scientific, Leicestershire, UK.

### 2.1. Methods

#### Preparation of Hyaluosomes

DCN-loaded hyaluosomes were prepared according to the thin film hydration technique using soya lecithin (200 mg) and Tween 80 (28.2 mg) as an edge activator (in the molar ratio of 85:15) with DCN. Briefly, specified amounts of soya lecithin, Tween 80 with DCN were dissolved in a 10 mL of chloroform: methanol mixture (7:3 *v*/*v*) in a round bottom flask (500-mL volume capacity). The round bottom flask was rotary evaporated under vacuum (Rotavapor, Heidolph VV 2000, Burladingen, Germany) at a temperature of 50 °C for 15 min at 150 rpm to allow the evaporation of the organic solvents and formation of a thin lipid film on the wall of the flask. Following this, hydration of the thin lipid film was carried out by addition of the hydrating medium 10 mL of PBS (pH 7.4), containing different amounts of HA into the round bottom flask, then rotating without vacuum for 30 min at 150 rpm until a milky dispersion of DCN-loaded hyaluosomes was obtained.

### 2.2. Characterization of the Prepared DCN-Loaded Hyaluosomes

#### 2.2.1. Determination of DCN Entrapment Efficiency Percentage (EE%)

Determination of DCN EE% of DCN was done via estimating the free amount (unentrapped) of DCN indirectly in the dispersion media. Centrifugation of 1mL of the prepared hyaluosomes at 22,000 rpm for 1 h at 4 °C using a cooling centrifuge (Sigma 3–30 KS, Sigma Laborzentrifugen GmbH, Osterode am Harz, Germany) was done so as to separate the free DCN from the hyaluosomes. The obtained supernatant was taken, then diluted, and the free DCN concentration was analyzed spectrophotometrically (Shimadzu, model UV-1800 PC, Shimadzu Corp., Kyoto, Japan) by estimating the ultraviolet (UV) absorbance at λ_max_ 258 nm. Each result represents the mean of three measurments ± SD.

Drug EE% was determined according to the following equation:$$\mathrm{EE}\%=\frac{\mathrm{total}\;\mathrm{amount}\;\mathrm{of}\;\mathrm{DCN}\;-\;\mathrm{unentrapped}\;\mathrm{DCN}}{\mathrm{total}\;\mathrm{amount}\;\mathrm{of}\;\mathrm{DCN}}\times 100$$

#### 2.2.2. Determination of Particle Size (PS), Zeta Potential (ZP), and Polydispersity Index (PDI)

Determination of the average PS, ZP, and PDI of the prepared hyaluosomes was carried out using the technique of dynamic light scattering by Zetasizer Nano ZS (Malvern Instrument Ltd., Worcestershire, UK). Each of the prepared hyaluosomes formulation was measured after being properly diluted in order to result in suitable intensity of scattering. ZP measuring was also done by utilizing the same equipment to notice the particles’ electrophoretic movement in the electric field. Each sample was measured for three times and the results represent the average value ± SD.

#### 2.2.3. Optimization of Hyaluosomes Using Factorial Design

Different formulations of DCN-loaded hyaluosomes were formulated according to 3^2^ full factorial experimental design to assess the effect of formulation independent variables on the EE%, PS, ZP and PDI of the prepared DCN-loaded hyaluosomes. Two variables were evaluated, each at 3 levels, namely: concentration of HA, and the drug concentration, each at three levels. The design was performed at all possible 9 combinations. The composition of different combinations prepared is presented in Table 1. Finally, Design expert^®^ 7 was used to assess the results and choose the optimized formulation.

#### 2.2.4. Evaluation of In Vitro Drug Release from Optimized Hyaluosomes System

DCN release from the optimized DCN-loaded hyaluosomes and DCN aqueous dispersion (2mg/mL) was carried via the technique of dialysis bag diffusion in a thermostatically controlled water bath shaker kept at 37 ± 0.5 °C. Before the experiment, the cellulose dialysis membrane was placed overnight in the release medium (PBS, pH = 7.4). One mL of the optimized DCN-loaded hyaluosomes or DCN aqueous dispersion (containing 2 mg DCN) were placed in the dialysis bag then tied at both ends. The dialysis tube was immersed in a bottle containing 20 mL of PBS (pH = 7.4) and placed in a thermostatically controlled shaker (Memmert, Munich, Germany) adjusted at 50 strokes per minute and 37 °C. Replacing the release medium with an equal volume of fresh medium was done at predetermined time intervals, in order to keep sink conditions, then measured spectrophotometrically ((UV-1800 PC), Shimadzu, Kyoto, Japan.) at the predetermined λ_max_ (258 nm) on the basis of standard curve previously constructed. All experiments were performed in triplicate.

#### 2.2.5. Transmission Electron Microscopy (TEM)

The morphology of the optimized DCN-loaded hyaluosomes and DCN-loaded liposomes without HA was visualized using TEM in order to show the spherical shape of vesicles and the gel core (Joel JEM 1230, Tokyo, Japan). A drop of the undiluted dispersion was spread on a carbon coated copper grid, then stained with a drop of 1% (*w*/*v*) phosphotungstic acid, air dried at room temperature, and finally examined at an accelerating voltage of 80 kV [24].

#### 2.2.6. Fourier Transform Infrared (FTIR) Spectroscopy

Before FTIR the optimized DCN-loaded and the corresponding blank hyaluosomes were stored in a freezer at −18 °C for 24 h, and then lyophilized in a freeze dryer (Novalyphe- NL 500; Savant Instruments Inc., Holbrook, NY, USA) under a temperature of −45 °C and vacuum of 7 × 10^−2^ mbar for 24 h.

FT-IR spectra between 4000 and 400 cm^−1^ (IRAffinity- 1; Shimadzu, Kyoto, Japan) of DCN, Soya lecithin, HA, and the optimized lyophilized DCN-loaded and blank hyaluosomes formulations were determined using potassium bromide (KBr) disc technique. Samples were crushed, combined together in a mortar with potassium bromide for 3–5 min, then compressed into disc by applying a pressure of 5 tons for 5 min in hydraulic press. The sample concentration in potassium bromide should be within the range of 0.2% to 1%. The pellets were placed into the path of light and the resulted spectrum was reviewed and checking for the presence of any chemical interactions [25].

#### 2.2.7. Differential Scanning Calorimetry (DSC)

The thermal analysis of DCN, Soya lecithin, HA, and the optimized lyophilized DCN-loaded and blank hyaluosomes formulations were performed using (Shimadzu differential scanning calorimetry, DSC- 60). Samples were placed in a pan formed of aluminum with a flat bottom and heated at a constant rate, using nitrogen as carrier gas to obtain dry atmosphere, in a range of temperature from 20 °C to 400 °C. The calibration of the temperature and energy scales of the instrument was carried using purified indium (99.9%) as the standard reference material.

#### 2.2.8. Sterilization by Gamma Radiation

The sterilization of the prepared DCN-loaded optimized hyaluosomes was done utilizing gamma radiation (^60^Co irradiator), (National Center for Radiation Research and Technology, Cairo, Egypt) at a dose of 10 kGy (1.19 kGy/hr). The selected optimized formulations were placed inside a polyurethane vessel and located in dry ice in to prevent accumulation or melting that may occur while carrying the sterilization process, when the temperature increased, by γ-irradiation. The sterilized sample retested for size, Zeta, and EE% after gamma sterilization.

### 2.3. In Vivo Arthritis Assessment in Rats

Experimental procedures were accepted by the Research Ethics Committee of Faculty of Pharmacy Cairo University (PI 2847) and followed the guide of the National Institute of Health for care and use of Laboratory animals (NIH Publications No.8023, revised 1978).

Thirty male Albino rats (250–300 g) were allocated in this study and placed in an airconditioned room at 25 ± 0.5 °C, standard pellet diet and water were freely accessed. The animals were supplied with day/night cycles of 12 h, after being accommodated in cages.

Knee joints of rats were injected two times separated by one-week (at days 1 and 7) with 0.5 mL of ovalbumin (5 mg/mL) in complete Freund’s adjuvant in order to induce OA [26]. Rats were allocated into 5 groups; each group contain 6 rats. Groups 1 and 2 were used for comparison, a negative control rats group having healthy knee joints and a positive control group in which no treatment was injected, respectively. Groups 3, 4, and 5 received 0.5 mL of IA injection of the blank hyaluosomes, the DCN aqueous dispersion (2mg/mL), and the optimized DCN-loaded hyaluosomes (F7) (2mg/mL), respectively. The left knee joint was kept free of treatment in all groups to be used as control [27].

Blood samples were taken from the rats’ retro-orbital plexus at different time intervals (days 14, 21 and 28) and placed in tube with gel barrier serum separator, first, before getting the first dose (day 14), the second withdrawal was after the first dose with one week (day 21) and the third withdrawal was after the second dose with one week (day 28), then the samples were centrifuged for 15 min at ambient temperature at 5000 rpm. The serum was frozen at 0 °C until analysis.

In order to determine the diameters of rats’ knees a micrometer (KM-211-101, Shaanxi, China) was used to measure the antero-posterior diameters at various time intervals (days 1, 7, 14, 21 and 28) to assess the treatment efficacy on the rats’ knee joints swelling [28]. The knee diameter baseline value was assessed just before the injection of adjuvant for each rat. The tumor necrosis factor-alpha (TNF-α) and interleukin 1-beta (IL-1b) serum levels were detected using an ELISA kit following the manufacturer’s instructions (Glory Science Co, Ltd., Del Rio, TX, USA). Cross-reactivity with other rat cytokines was excluded, according to the manufacturer [29].

### 2.4. Histopathologic Studies

At day 28, animals were sedated and euthanized via intravenous delivery of sodium pentobarbital (50–80 mg/kg), then both knees were isolated.

The samples of the knee joints were placed in 10% neutral buffered formalin for 48h, then decalcified using Cal-Ex II Decalcifier for 18 days. Serial dilutions of ethanol were prepared for the processing of samples, then they were placed in xylene to be cleared, finally they were embedded in paraplast tissue embedding media. Tissue sections (5 µm) were cut by rotatory microtome, serial sagittal tissue sections were cut at mid joint levels and mounted on glass slides to be stained with hematoxylin and eosin (H&E) staining for general tissue histological examination then imaged by Full HD microscopic imaging system (Leica Microsystems GmbH, Wetzlar, Germany) [30]. Morphological examination of tibiofemoral articular cartilage were carried out with referring to modified Mankin scoring system [31,32] where 0 indicating normal cartilage while 13 indicate the maximal score of OA.

## 3. Results and Discussion

DCN-loaded hyaluosomes were successfully prepared by the thin film hydration technique. Design expert^®^ 7 software program (version 7, Stat-Ease, Minneapolis, MN, USA) was used to statistically analyze the factorial design and test the significance of different formulation variables on the EE%, PS, ZP, and PDI

### 3.1. Results of Entrapment Efficiency Percentage EE%, Particle Size (PS), Zeta Potential (ZP), and Polydispersity Index (PDI)

Table 1 shows the results of the EE%, PS, ZP, and PDI of different DCN-loaded hyaluosomes. All formulations showed reasonable EE% (from 60% to 93%). Additionally, they succeeded to achieve nanosized particles (PS range from 300 to 450 nm) with good size distribution (PDI from 0.2 to 0.4) and a negative ZP (from −11 to −14.5 mv).

ANOVA analysis was used to evaluate the level of significance of the tested variables on the EE%, PS, ZP and PDI of the prepared hyaluosomes. ANOVA results showed that the drug concentration and the HA concentration had significant effect on the EE% of DCN-loaded hyaluosomes (*p* < 0.0001) and (*p* = 0.0067), respectively. Figure 1a shows the main effects of the drug concentration and HA concentration on the EE%, where increasing the drug amount results in more drug encapsulation, thus increasing the EE% [33]. This could be attributed to the hydrophobic nature of the drug. Similar results were obtained by Padamwar et al. [34] who stated that high EE% may be attributed to high lipophilicity of the drug. HA also had a significant positive effect on the drug EE%, where increasing its concentration resulted in significantly higher DCN EE%. This may be attributed to the increased steric stabilization due to presence of HA long chains with repeated units of sugar that can stick out of the vesicles and sterically hinder the escape of the drug from the prepared hyaluosomes, therefore increases the EE% [35]. Similar results were obtained by Kawar et al. [22] in their study on the addition of HA to liposomes, where they found that HA resulted in the production of more stable vesicles with lower chance to aggregate as the formed hyaluosomes have lower tendency to leak the entrapped drug molecules.

None of the tested variables had a significant effect on the PS or The PDI of the prepared hyaluosomes (*p* > 0.05). Concerning the ZP, all the prepared hyaluosomes were negatively charged, owing to the presence of Soya lecithin [36] and HA which carry negative surface charge [21]. ANOVA results revealed that only the HA concentration has significant positive impact (*p* = 0.0128). Figure 1b showed that increasing HA concentration results in the formation of vesicles with significantly higher ZP. This is attributed to the negative charge of the HA molecule [37,38].

### 3.2. Statistical Optimization of the Results

Design expert^®^7 was used to choose the optimized formulation that could achieve the highest EE%, lowest PS, ZP values in range (−11 to −14.5), and PDI values in range (0.2 to 0.4). Optimization results revealed that DCN-loaded hyaluosomes system (F7), prepared using 0.2% HA and loaded with 20 mg/10 mL DCN is the optimum formulation with the highest desirability factor (0.804). This optimized formulation (F7) was prepared and re-evaluated, and the results of the predicted and actual responses are presented in Table 2. The difference between the predicted and estimated values calculated using the following equation:$$\%\;\mathrm{Difference}=\frac{\mathrm{predicted}\;\mathrm{value}-\mathrm{estimated}\;\mathrm{value}}{\mathrm{predicted}\;\mathrm{value}}\times 100$$

All estimated results were found to be within 10% difference from the predicted value, indicating that the design is reliable in estimating the actual responses values.

### 3.3. Result of In Vitro Drug Release from the Optimized Hyaluosomes Formulation

The percentages DCN released from optimized DCN-loaded hyaluosomes and DCN dispersion in distilled water at different times intervals were measured, and the release profiles are graphically illustrated in Figure 2. Results showed that the optimized formulation (F7) succeeded to sustain the release of DCN compared to that from DCN aqueous dispersion. The optimized DCN-loaded hyaluosomes (F7) released less than 50% DCN over 48 h, whereas more than 50% were released in the first 4 h from the aqueous DCN dispersion. Nanovesicles containing soya lecithin forming the lipid bilayer can help in retarding diffusion of dissolution media inside the hyaluosomes, resulting in sustained DCN release [39]. In addition, HA has a bulky structure with long chains of sugar groups is supposed to remain in the core of the phospholipid bilayer vesicles, although these repeated long chains of sugar units can stick out of the vesicles leading to steric stabilization which in turn resulted in sustaining the release of DCN from the hyaluosomes [22]. Similar results were obtained by Shao et al. [40] in their study on a HA modified, doxorubicin co-delivered hybrid nano system for leukemia therapy, where they found that HA coating on the nano system delayed the drug release.

### 3.4. Results of Transmission Electron Microscopy (TEM)

TEM micrographs of the optimized DCN-loaded hyaluosomes and DCN-loaded liposomes (without HA) revealed vesicles that are spherical in shape, yet non-aggregating, having smooth surface and sharp boundaries as illustrated in Figure 3a,b. Moreover, the observed diameter by TEM micrographs was in accordance with the size obtained by the Zetasizer. Figure 3b showed the gel core hyaluosomes in which HA appears in the hydrophilic core of the prepared hyaluosomes. Similar results were obtained by described by El-Refaie et al. [21] in their study on Gel-core Hyaluosomes for transdermal drug delivery of HA.

### 3.5. FTIR Results

FT-IR spectra were obtained for individual components (Soya Lecithin, HA, DCN) as well as optimized lyophilized formulation (F7) (Figure 4a–e). The IR chart of soy lecithin (Figure 4a) showed the alkane bands corresponding to symmetric CH_2_ at 2854 cm^−1^, the asymmetric CH_2_ at 2928 cm^−1^, the asymmetric CH_3_ stretching and the CH_2_ scissoring vibrational modes at 2956 and 1462 cm^−1^, respectively. the carbonyl stretching vibration, located at 1736 cm^−1^, and the highly overlapped PO_2_^−^and P–O–C infrared active vibrations in the region between 1200 and 970 cm^−1^, centered around 1061 cm^−1^ [41].

The IR chart of HA (Figure 4b) shows a band at 3441 cm^−1^ (corresponding to the OH and NH stretching region), a band at 2897 cm^−1^ (corresponding to stretching vibration of C-H), a band at about 1647 cm^−1^ (corresponding to the amide carbonyl) and at 1415 cm^−1^ (corresponding to the stretching of COO^−^), which refers to the acid group of molecule HA. The absorption band at 1045 cm^−1^ is attributed to the linkage stretching of C-OH [42].

The IR chart of DCN (Figure 4c) showed characteristic bands at 3329 cm^−1^ (corresponding to the stretching of –OH stretch), 3070 cm^−1^ (corresponding to the stretching of aromatic C–H), 2939 cm^−1^ (corresponding to the stretching of aliphatic C–H), 1770 cm^−1^ (corresponding to the ester C=O), 1678 cm^−1^ (corresponding to the stretching of C=O group of COOH), 1593.20 cm^−1^ (corresponding to the stretching of aromatic C=C), 1450.47 cm^−1^ (corresponding to the stretching of COO^−)^, 1026 cm^−1^ (corresponding to the stretching of ester C–O), 744.52 cm^−1^ and 705.96 cm^−1^ (corresponding to the m-substituted benzene and benzene, respectively) [43].

The IR chart of blank formulation (Figure 4d) did not show any shifts in the bands of the individual components. The FT-IR spectrum for the optimized DCN-loaded hyaluosomes formulation (F7) (Figure 4e) did not show any shifts in the bands of both the drug and the individual components. Thus, it can be concluded that based on FT-IR spectra, there is no chemical interaction between the drug and the individual components used in the preparation of the optimized formulation.

### 3.6. Results of Differential Scanning Calorimetry (DSC)

DSC is a common tool to analyze the physical properties of the material and to give information onto the melting and recrystallization manner of crystalline materials [44]. Figure 5 shows the DSC thermograms for DCN, soy lecithin, HA, and the optimized lyophilized DCN-loaded and the corresponding blank hyaluosomes. The DSC thermogram of DCN showed a sharp endothermic peak at about 255 °C corresponding to its melting point [45]. The DSC thermogram of Soya Lecithin showed an endothermic peak around 155 °C due to its melting point [46]. The DSC thermogram of the HA showed an exothermic peak at around 240 °C that can be attributed to the degradation of the polysaccharide [47]. No new peaks were observed in the thermogram of the blank hyaluosomes, indicating the absence of any unwanted interactions. Only the intensity of peaks has been decreased due to the dilution occurring during formulation. Total disappearance of DCN peak was noticed in the thermogram of the DCN-loaded hyaluosomes (F7) confirming that DCN was molecularly dispersed or present in the amorphous state within the hyaluosomes. Similar results were obtained in several studies in the literature [48,49,50].

### 3.7. Results of Sterilization by Gamma Radiation on the Stability of the Optimized Hyaluosomes

PS, ZP and EE% were determined before and after sterilization to evaluate the effect of the sterilization process on the properties of the vesicles. Results confirmed that sterilization process had no significant impact on the prepared DCN-loaded hyaluosomes, where the drug EE%, PS, and ZP were not significantly changed after sterilization. The PS of the sterilized optimized hyaluosomes formulation (F7) was not significantly different before and after sterilization (310, 300 nm before and after sterilization, respectively), ZP was not significantly different before and after sterilization (12.2, 12 mV before and after sterilization, respectively), also the % EE was not significantly different after sterilization (90.7%, 90% before and after sterilization, respectively) indicating the stability of the hyaluosomes formulation after exposure to gamma radiation at the dose of 10 KGy.

### 3.8. Results of In Vivo Study

An encouraging treatment for OA is expected to relief pain and inflammation and to keep healthy joints. The aim of this study was to evaluate the remedial effects shown after treatment with DCN together with HA, such as enhancement in the inflammatory condition, and deterioration of cartilage in rats that were injected with antigen to induce arthritis. The formulated injections were: the blank hyaluosomes, DCN aqueous dispersion (2mg/mL), and the optimized DCN-loaded hyaluosomes (F7) (2 mg/mL), respectively. Each of them was injected IA at the knee of rats.

As a main pharmacodynamic evaluation of the arthritis progression and administered treatment efficacy, the rats’ knee swelling was assessed by determining its diameter. The swelling of the knee was defined as the change between the recorded value at normal condition before arthritis induction and the value measured after arthritis induction, at different times. The results are presented in Figure 6a.

After one and two weeks of antigen injection (at days 7 and 14, respectively), all the groups revealed significant swelling of joint relative to the negative control group (*p* < 0.05). After one week from the beginning of the treatment (on day 21), the swelling of the knee was significantly decreased, (*p* < 0.05). Moreover, the swelling of knee among all the groups that were received treatments was significantly lower than the group that was left without treatment (positive control group) (*p* < 0.05), but the enhancement was more significant in the selected DCN-loaded hyaluosomes injection (F7).

The anti-edematous activity of the treatment that was injected intra-aricularly clearly appeared after receiving the second dose of treatment by one week (at day 28), where the swelling of the knee of the treated groups remained significantly lower than the untreated groups; in addition, the differentiation between the various treated groups became more obvious. Such that the degree of knee swelling was in the following order: group 5 (receiving the selected DCN-loaded hyaluosomes (F7)) < group 4 (receiving DCN aqueous dispersion) < group 3 (receiving blank hyaluosomes), (*p* < 0.05).

The results were harmonized with our objectives and through the light on the potential favorable effect of the selected formulation F7. Although, the plasma tumor necrosis factor-alpha (TNF-α) and interleukin 1-beta (IL-1b) which are the main proinflammatory cytokines, are greatly produced in the affected joint and contribute to the pathogenesis of OA. These cytokines further induce the production of other inflammatory mediators such as: some prostaglandins and cycloxygenase-2 from synovial cells which induce more inflammation and cartilage deterioration [51]. Serum levels of TNF-α and IL-1b are highly augmented in such inflammatory milieu which are suppressed by the proper use of anti-inflammatory drugs and they are considered good markers for assessment of pharmacological efficacy and management of synovial inflammatory disorders of our new formulation [52].

Table 3 and Figure 6b,c show the results of the rat plasma TNF-α and IL-1b levels measured after 4 weeks of OA induction. Their levels were elevated significantly in the arthritic group (group 2) as compared to their levels in the groups receiving treatment (groups 3, 4, and 5). A considerable synchronization, between the swelling of the knee results, and TNF-α and IL-1b plasma levels can be noticed. It is clear that the reduction in TNF-α levels IL-1b levels were in the following order: group 5 (receiving the selected DCN-loaded hyaluosomes (F7)) < group 4 (receiving DCN aqueous dispersion) < group 3 (receiving blank hyaluosomes), (*p* < 0.05). These results are in harmonization with our anticipation regarding the supremacy of the optimized formulation F7 over the blank hyaluosomes or DCN aqueous dispersion, through sustaining the release of DCN, and consequently, prolonging its therapeutic effect, together with the HA chondrogenic effect which lead to a synergistic effect with DCN.

### 3.9. Histopathological Evaluation

The inflammation degree of tibiofemoral articular cartilage was evaluated using modified Mankin score, Figure 7, together with, microscopic histopathological investigation using both H&E stains Figure 8a–e. Referred to modified Mankin grading, a score was given to inflammation starting from 0 to 13, where, 0 refers to normal cartilage, while 13 is the maximal score, for osteoarthritic model. Different study groups revealed significant difference (*p* < 0.05).

Gp.1 (negative control of rats with healthy knee joints) demonstrated normal histological features of covering hyaline cartilages with intact smooth articular surfaces, well organized apparent intact chondrocytes all over cartilaginous zones with large vesicular intact nuclei (Arrows). Intact synovial membranes were observed with minimal inflammatory cells infiltrates records and normal vasculatures (Figure 8a). (Mean Mankin score = 0.5).

Gp.2 (positive control of rats receiving no treatment) samples showed wide areas of cartilaginous surface erosions and fissures (star) with significant loss of chondrocytes and many degenerated and necrotic changes records (red arrow) accompanied with congested subchondral blood vessels (dashed arrow). Abundant inflammatory cells infiltrate records in synovial membranes (red arrow—Figure 8b) with mild focal hyperplasia of covering epithelium. (Mean Mankin score = 10).

Gp.3 (receiving Blank hyaluosomes) samples showed persistence of minor focal articular surface erosions (star) and moderate degenerative changes of chondrocytes (red arrow). Milder persistence records of inflammatory cells infiltrate records of synovial membranes were observed (Figure 8c Red arrow). (Mean Mankin score = 7).

Gp.4 (receiving DCN aqueous dispersion (2mg/mL)) samples demonstrated almost intact, smooth articular surfaces without abnormal alterations; many apparent intact chondrocytes were observed (black arrow) with focal areas of degenerative changes and pyknotic nuclei in deeper cartilaginous zones (red arrow). Synovial membranes showed few occasional inflammatory cells infiltrates (Figure 8d Red arrow) with many congested blood vessels (star). (Mean Mankin score = 4).

Gp.5 (receiving the optimized DCN-loaded hyaluosomes, F7) showed well organized morphological features of articular cartilage with many apparent intact chondrocytes (black arrow) and minimal records of degenerative changes (red arrow). Almost intact synovial membranes were observed with occasional subepithelial focal inflammatory cells infiltrates (Figure 8e Red arrow) and fewer congested blood vessels (star). (Mean Mankin score = 3).

These data prove that DCN-loaded hyaluosomes can obviously enhance the results of the knee swelling, plasma TNF-α and IL-1b levels. This indicates the significant chondrogenic effect of the optimized DCN-loaded hyaluosomes.

Results of knee swelling together with the plasma levels of TNF-α and IL-1b were correlated to each other. This result suggests that DCN-loaded hyaluosomes promote great improvement in OA inflammation, as together HA and DCN succeed in reducing inflammation, allowing tissue repair as well.

## 4. Conclusions

In the present study, intra-articular injectable hyaluosomes using Soy lecithin, loaded with a structural modifying OA drug (DCN), and natural component of articular cartilage and synovial fluid (HA) was successfully formulated and evaluated for their particle size (358 nm), zeta potential (13.6 Mv), entrapment efficiency (87%) and in vitro drug release. The obtained results revealed that using 0.2% (*w*/*v*) HA loaded with 20 mg/10mL DCN, produced hyaluosomes with the most sustained drug release profile (48% at 48 h) and highest reduction of inflammation. The presence of HA clearly enhanced the knee swelling in rats. Thus, this DCN-loaded hyaluosomes could be used as suitable alternative for DCN oral treatment.

## Figures and Tables

**Figure 1 pharmaceutics-13-00765-f001:**
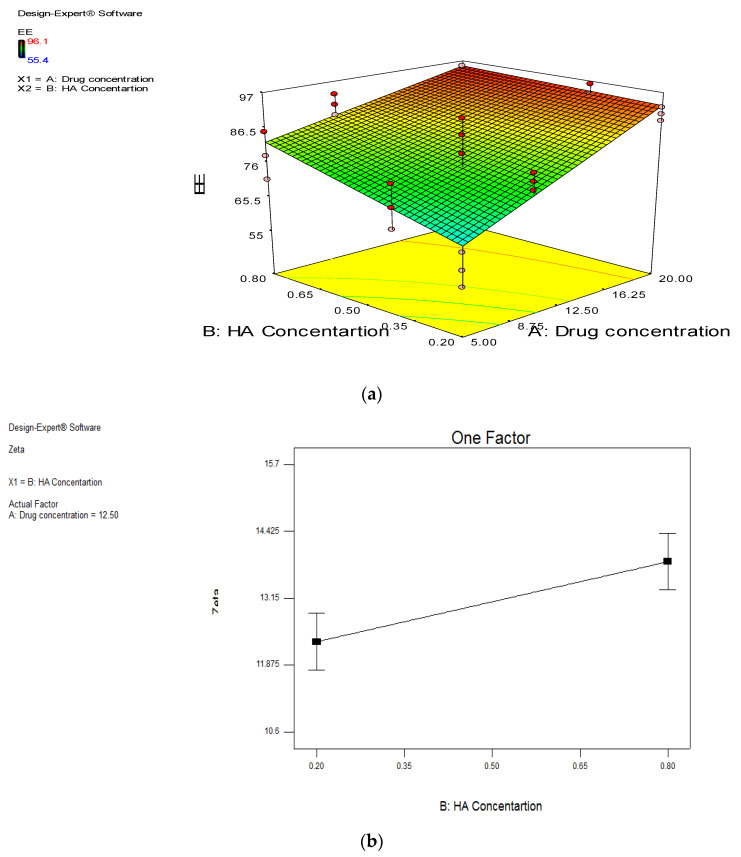
Chart showing: (**a**) Effect of Drug concentration on EE% and effect of HA concentration on the EE%; (**b**) Effect of HA concentration on ZP.

**Figure 2 pharmaceutics-13-00765-f002:**
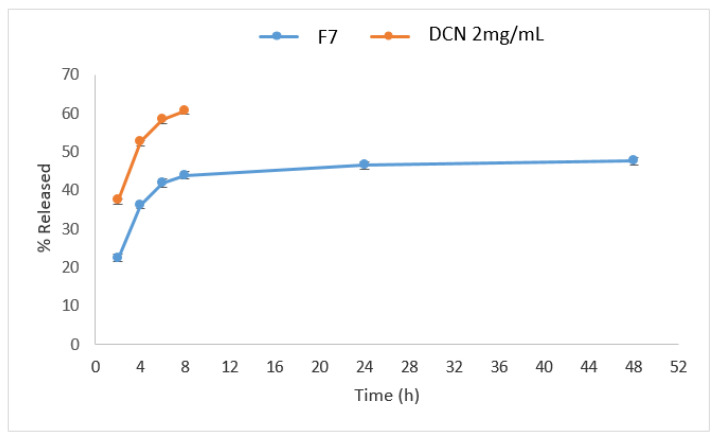
Release profile of DCN from the optimized hyaluosomes (F7) and DCN aqueous dispersion (2mg/mL).

**Figure 3 pharmaceutics-13-00765-f003:**
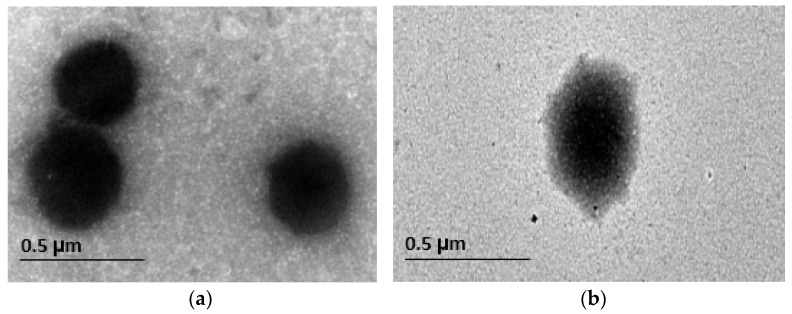
Transmission electron micrographs of: (**a**) DCN-loaded vesicles (without HA); (**b**) optimized DCN-loaded hyaluosome (with gel core).

**Figure 4 pharmaceutics-13-00765-f004:**
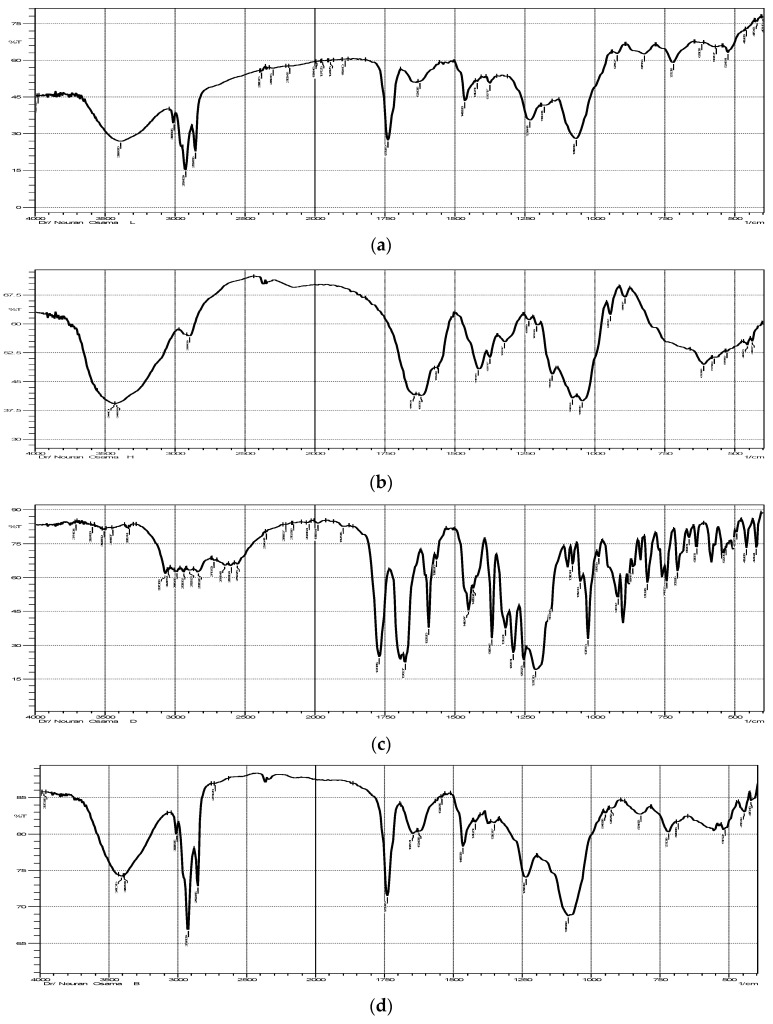
FT- IR spectra of Soya lecithin (**a**), HA (**b**), DCN (**c**), Blank formulation (**d**), and the optimized hyaluosome formulation (F7) (**e**).

**Figure 5 pharmaceutics-13-00765-f005:**
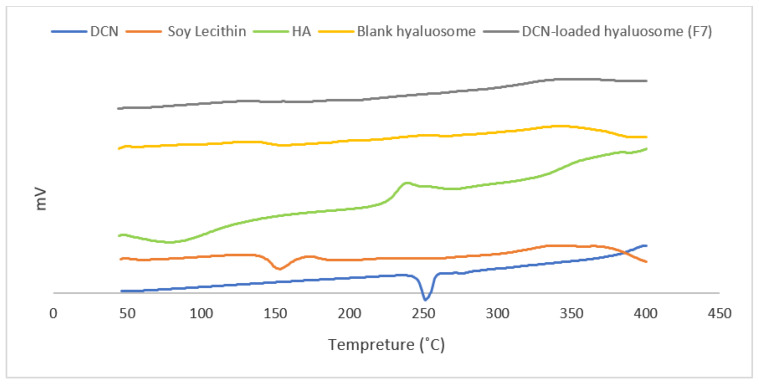
DSC thermograms for DCN, Soya lecithin, HA, and the optimized lyophilized DCN-loaded (F7) and blank hyaluosome formulation.

**Figure 6 pharmaceutics-13-00765-f006:**
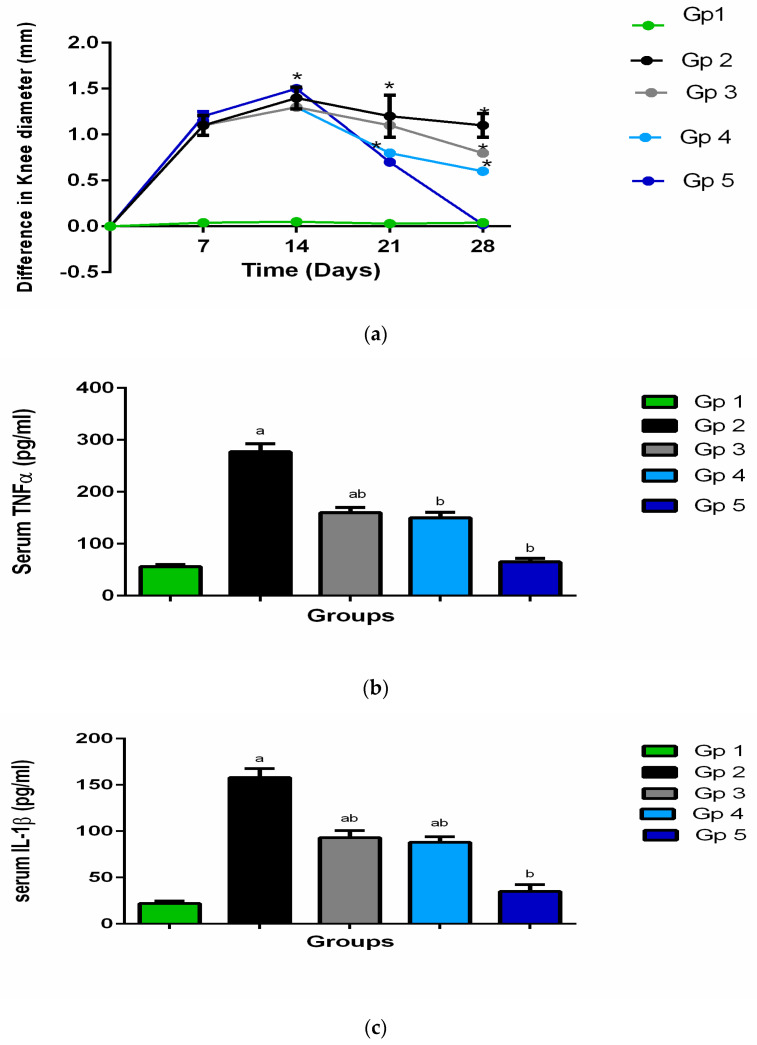
(**a**) The knee swelling for GP1 (negative control), GP2 (positive control), GP3 (Blank hyaluosome), GP4 (DCN aqueous dispersion), and GP5 (optimized DCN loaded hyaluosomes (F7)); (**b**) The rat’s plasma TNF-α levels; (**c**) The rat’s plasma IL-1b levels. ^a^ Significantly different from negative control group at *p* < 0.05, * ^or b^ Significantly different from positive control group at *p* < 0.05, ^ab^ Significantly different from both negative and positive control group at *p* < 0.05.

**Figure 7 pharmaceutics-13-00765-f007:**
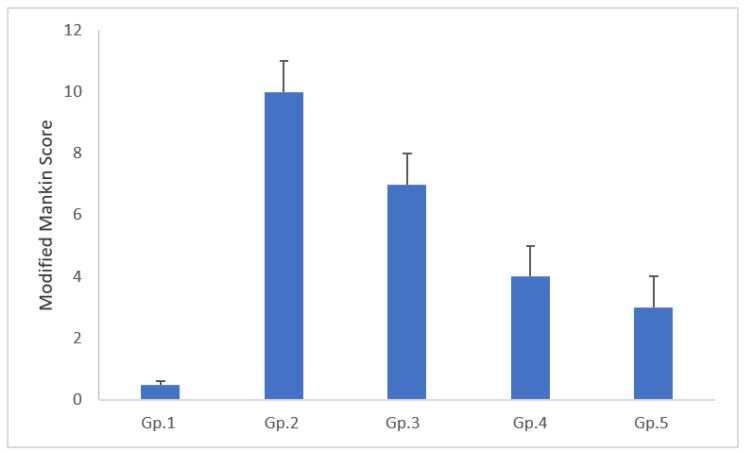
Morphological assessment of articular cartilage of different study groups: GP1 (negative control), GP2 (positive control), GP3 (Blank hyaluosome), GP4 (DCN aqueous dispersion), and GP5 (optimized DCN loaded hyaluosomes (F7)) using modified Mankin score.

**Figure 8 pharmaceutics-13-00765-f008:**
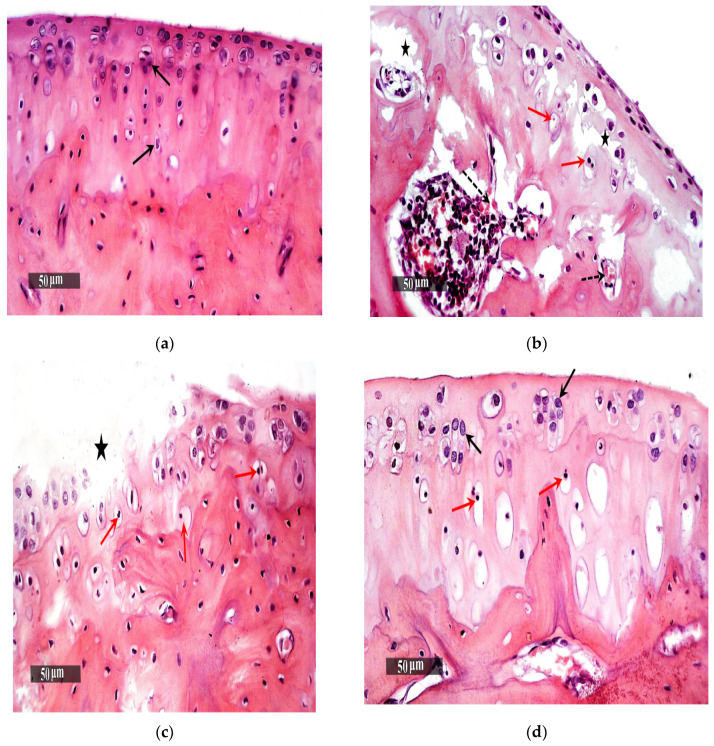
Morphological features of (**a**) GP.1 negative control, (**b**) GP.2 positive control, (**c**) GP.3 Blank hyaluosome, (star represent the presence of minor focal articular surface erosions), (**d**) DCN aqueous dispersion, and (**e**) GP.5 Optimized DCN-loaded hyaluosome (F7).

**Table 1 pharmaceutics-13-00765-t001:** Composition of different hyaluosomes formulations.

Formulations *	Drug Concentration (mg/10 mL)	Hydration Media	EE% *	PS (nm) *	ZP * (mV)	PDI *
F1	5	0.2% HA	60.2 ± 5	330 ± 2.5	−12 ± 1.1	0.4 ± 0.0
F2		0.4% HA	72.4 ± 6.8	370 ± 7.2	−13.5 ± 1.3	0.22 ± 0.1
F3		0.8% HA	78.2 ± 7.2	336 ± 2.3	−14 ± 1.0	0.27 ± 0.0
F4	10	0.2% HA	79.8 ± 2.5	309 ± 3.0	−11.6 ± 0.9	0.30 ± 0.0
F5		0.4% HA	88.5 ± 4.8	393 ± 3.9	−11.8 ± 0.8	0.33 ± 0.1
F6		0.8% HA	89.8 ± 3.2	342 ± 2.7	−13 ± 1.15	0.27 ± 0.0
F7	20	0.2% HA	90.7 ± 2	310 ± 2.1	−12.2 ± 0.6	0.36 ± 0.0
F8		0.4% HA	92.9 ± 3.2	420 ± 3.6	−14.5 ± 1.2	0.22 ± 0.1
F9		0.8% HA	93.7 ± 1.6	350 ± 1.8	−13 ± 1.0	0.35 ± 0.0

* Data are mean values (*n* = 3) ± SD.

**Table 2 pharmaceutics-13-00765-t002:** The predicted and estimated values of different responses of the optimized formulation (F7).

	EE%	PS (nm)	ZP (mV)
Predicted	93	350	12.6
Measured	87	358	13.6
% Difference	6.4	2.2	7.9

**Table 3 pharmaceutics-13-00765-t003:** Results of the rat’s plasma TNF-α and IL-1b levels measured after 4 weeks of osteoarthritis induction.

Groups	Inflammatory Markers * Serum Concentrations (pg/mL), (% Inhibition)	
	TNFα	IL-1b
GP.1 Negative control	56 ± 4.13	22 ± 2.67
GP.2 Positive control	277 ^a^ ± 15.34	158 ^a^ ± 9.67
GP.3 Blank hyaluosome	160 ^a,b^ ± 9.97 (42)	93 ^a,b^ ± 7.81 (41)
GP.4 DCN aqueous dispersion 2mg/ml	150 ^b^ ± 10.88 (45)	88 ^b^ ± 5.99 (44)
GP.5 DCN loaded hyaluosomes F7	65 ^b^ ± 6.93 (76)	35 ^b^ ± 7.32 (77)

After 4 weeks of osteoarthritis induction, respectively. * Data are represented as mean ± standard error (*n* = 6). Statistical significance was considered at *p* < 0.05 based on one-way analysis of variance ANOVA test followed by Tukey’s test for multiple comparisons. ^a^ Significantly different from negative control group at *p* < 0.05, ^b^ Significantly different from positive control group at *p* < 0.05.

## Data Availability

Not applicable.

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
