# Peer review of "Diacerein-Loaded Hyaluosomes as a Dual-Function Platform for Osteoarthritis Management via Intra-Articular Injection: In Vitro Characterization and In Vivo Assessment in a Rat Model"

_pharmaceutics, 2021, doi:10.3390/pharmaceutics13060765_

Round 1

Reviewer 1 Report

In this paper the Authors investigated the development of intra-articular injectable hyaluronic acid gel-core vesicles loaded with diacerein, for the treatment of osteoarthritis. The optimal formulation was designed with a full factorial design and characterized in terms of drug entrapment efficiency, particle size, zeta potential, DSC; FTIR and TEM. The prepared DCN-loaded hyaluosomes improve the in-vivo inflammatory condition and prevent cartilage damage,showing a good ability for the  management of osteoarthritis after intra-articular injection. 

Revision:

abstract: typing error: after lately insert a point. Why is diacerein reported with capital letter?

Please revise the phrase: the “swelling following intra-articular injection” since is not clear

Introduction 

Some references about diacerein formulation in colloidal systems should be cited , such as:

Aziz DE, Abdelbary AA, Elassasy AI. Fabrication of novel elastosomes for boosting the transdermal delivery of diacerein: statistical optimization, ex-vivo permeation, in-vivo skin deposition and pharmacokinetic assessment compared to oral formulation. Drug Deliv. 2018 Nov;25(1):815-826. doi: 10.1080/10717544.2018.1451572. PMID: 29557244; PMCID: PMC6058680.

El-Say KM, Abd-Allah FI, Lila AE, Hassan Ael-S, Kassem AE. Diacerein niosomal gel for topical delivery: development, in vitro and in vivo assessment. J Liposome Res. 2016;26(1):57-68. doi: 10.3109/08982104.2015.1029495. Epub 2015 Apr 8. PMID: 25853339. 

page 2- Please revise the phrase:”enhance the motion range” since is not clear

page 3 please report the acronymous EE, PS, and ZP

Material

cut off of Cellulose dialysis membrane should be reported

Which kind of hyaluronic acid sodium salt was employed? Mw?

Please report the used amount of soya lecithin and Tween 80

Results

Release studies:

Could the Authors report the solubility of diacerin in this condition? Since diacerein is not soluble in water (3.197 mg/L as reported in the Introduction)  it is reasonable that the 100% of drug can’t be released but why the detection was stopped at 8 hours  for while it was continued for the formulation?

Have the Author compared drug alone with purified hyalurosomes where entrapped drug was separated by unetrapped ?

Is diacerein stable in water for 48h?

Concerning the formulation: the remaining 50% of drug is in the core? The Authors could easily assess that disrupting the vesicles after the test and analyzing drug content.

The stability of diacerein in the formulation should be analyzed.

TEM:

I can’t appreciate “ the gel core hyaluosomes in which HA appears in the hydrophilic core of the prepared hyaluosomes” maybe with a Cryo-EM.

Author Response

On behalf of the authors, I would like to thank you and the reviewers for your time and efforts to improve the submitted work. All the reviewers’ suggestions and comments have been considered with great attention and the manuscript has been modified accordingly to provide a proper response to each comment. We hope that the manuscript in its current form will meet the criteria for publication in Pharmaceutics

Dr. Rehab Shamma

Response to reviewers’ comments

Reviewer # 1

In this paper the Authors investigated the development of intra-articular injectable hyaluronic acid gel-core vesicles loaded with diacerein, for the treatment of osteoarthritis. The optimal formulation was designed with a full factorial design and characterized in terms of drug entrapment efficiency, particle size, zeta potential, DSC; FTIR and TEM. The prepared DCN-loaded hyaluosomes improve the in-vivo inflammatory condition and prevent cartilage damage, showing a good ability for the management of osteoarthritis after intra-articular injection. 

Revision:

abstract: typing error: after lately insert a point. Why is diacerein reported with capital letter?

Response: The reviewer’s correction has been addressed.

Please revise the phrase: the “swelling following intra-articular injection” since is not clear

Response: The reviewer’s comment has been addressed and the phrase has been revised and rephrased.

Introduction 

Some references about diacerein formulation in colloidal systems should be cited, such as:

Aziz DE, Abdelbary AA, Elassasy AI. Fabrication of novel elastosomes for boosting the transdermal delivery of diacerein: statistical optimization, ex-vivo permeation, in-vivo skin deposition and pharmacokinetic assessment compared to oral formulation. Drug Deliv. 2018 Nov;25(1):815-826. doi: 10.1080/10717544.2018.1451572. PMID: 29557244; PMCID: PMC6058680.

El-Say KM, Abd-Allah FI, Lila AE, Hassan Ael-S, Kassem AE. Diacerein niosomal gel for topical delivery: development, in vitro and in vivo assessment. J Liposome Res. 2016;26(1):57-68. doi: 10.3109/08982104.2015.1029495. Epub 2015 Apr 8. PMID: 25853339. 

 Response: The reviewer’s suggestions have been followed and the references were added

page 2- Please revise the phrase:”enhance the motion range” since is not clear

Response: The reviewer’s comment has been addressed and the phrase has been replaced with the “enhance the flexibility”.

page 3 please report the acronymous EE, PS, and ZP

Response: The recommendation of the reviewer is appreciated. The acronymous have been reported.

Material

cut off of Cellulose dialysis membrane should be reported

Response: The recommendation of the reviewer is appreciated. cut off of Cellulose dialysis membrane has been reported.

Which kind of hyaluronic acid sodium salt was employed? Mw?

Response: The used type is sodium hyaluronate,95% was purchased from Acros Organics Co., with Molecular weight 1.5 to 2.2 million Da. This has been added to the text.

Please report the used amount of soya lecithin and Tween 80

Response: The recommendation of the reviewer is appreciated. The used amounts of soya lecithin and Tween 80 have been mentioned.

Results

Release studies:

Could the Authors report the solubility of diacerin in this condition? Since diacerein is not soluble in water (3.197 mg/L as reported in the Introduction), it is reasonable that the 100% of drug can’t be released but why the detection was stopped at 8 hours  for while it was continued for the formulation?

Response: The release medium used in this study is PBS( pH 7.4) in which diacerein has high solubility as stated by Pimple et al (2014), where they reported that diacerein solubility in PBS pH 7.4 is 0.599 mg/ml, which is 3 fold higher than its water solubility (0.188 mg/ml).

Moreover, we have measured diacerein solubility in PBS pH 7.4 as per your request and found to be 1 mg/ml, therefore the drug can be released completely in the used release medium.

  • Pimple, S., P. Maurya, A. Joshi, A. Swami, and R. Singh, formulation development and evaluation of diacerein hard gelatin capsules. IJPRD, 2014. 6(8): P.19 – 25.

The used release medium was previously used for diacerein release from diacerein loaded nanovesicles in studies preformed by Aziz et al (2018, 2019).

  • Aziz DE, Abdelbary AA, Elassasy AI. Fabrication of novel elastosomes for boosting the transdermal delivery of diacerein: statistical optimization, ex-vivo permeation, in-vivo skin deposition and pharmacokinetic assessment compared to oral formulation. Drug Deliv. 2018 Nov;25(1):815-826
  • Aziz DE, Abdelbary AA, Elassasy AI. Investigating superiority of novel bilosomes over niosomes in the transdermal delivery of diacerein: in vitro characterization, ex vivo permeation and in vivo skin deposition study. Journal of Liposome Research. Volume 29, 2019 - Issue 1.

The detection stopped for the drug at 8 hours as more than 50% was released after 4 h and this prove the rapid release of the drug aqueous dispersion compared to sustained release of our selected formulation.

Have the Author compared drug alone with purified hyalurosomes where entrapped drug was separated by un-etrapped?

Response:  The formulation prepared is intended to be used as a whole, without separation of the vesicles. The unentrapped part of drug gives initial burst release, which is beneficial, then the rest of entrapped drug will be released over a prolonged period of time to ensure sustainment of the effect.  

Is diacerein stable in water for 48h?

Response: A previous study was done to check the stability of diacerein in aqueous media. Hamrapurkar et al (2011) performed a stability indicating assay for diacerein, and reported no significant changes in solution stability, and confirmed that the sample solutions were stable for at least seven days.

Hamrapurkar, P., P. Patil, M. Desai, M. Phale, and S. Pawar, Stress degradation studies and development of a validated stability-indicating-assay-method for determination of diacerein in presence of degradation products. Pharmaceutical methods, 2011. 2(1): p. 30-35,

Concerning the formulation: the remaining 50% of drug is in the core? The Authors could easily assess that disrupting the vesicles after the test and analyzing drug content.

Response: The recommendation of the reviewer is appreciated and the drug content after disrupting the vesicles at the end of the release after 48 h was measured, and found to be 47%

The stability of diacerein in the formulation should be analyzed.

Response: The recommendation of the reviewer is appreciated.

Previous studies reported the stability of diacerein in nano systems formulation.

Parekh et al (2017) prepared diacerein nanosuspension and evaluated its stability. Their results confirmed that the formulation was physically and chemically stable when stored at the 40 ± 2°C and 75 ± 5 % RH for a period of one month.

  • Parekh, K.K., J.S. Paun, and M.M. Soniwala, Formulation and evaluation of nanosuspension to improve solubility and dissolution of diacerein. International Journal of Pharmaceutical Sciences and Research, 2017. 8(4): p. 1643.

In addition, diacerein loaded nanovesicles were previously prepared in several studies.

  • Aziz DE, Abdelbary AA, Elassasy AI. Fabrication of novel elastosomes for boosting the transdermal delivery of diacerein: statistical optimization, ex-vivo permeation, in-vivo skin deposition and pharmacokinetic assessment compared to oral formulation. Drug Deliv. 2018 Nov;25(1):815-826
  • Aziz DE, Abdelbary AA, Elassasy AI. Investigating superiority of novel bilosomes over niosomes in the transdermal delivery of diacerein: in vitro characterization, ex vivo permeation and in vivo skin deposition study. Journal of Liposome Research. Volume 29, 2019 - Issue 1.

TEM:

I can’t appreciate “ the gel core hyaluosomes in which HA appears in the hydrophilic core of the prepared hyaluosomes” maybe with a Cryo-EM.

Response: The recommendation of the reviewer is appreciated. Unfortunatley Cryo-EM is not available at my institute.

Reviewer 2 Report

Although the work is interesting and well structured, it is necessary to correct the English language because many sentences are not clear (e.g. Although these drugs are recommended by the societies of rheumatology societies and government agencies as they are considered as symptom managing rapid-acting drugs in OA and, treatment of the actual cause of pain remains neglected –page 1; HA (also known as hyaluronan) is utilized in joint diseases treatment, such as OA and rheumatoid arthritis [13], where in these disorders, both the concentration and molecular weight of HA in the synovial fluid have declined and the synovial fluid become less elastic and less viscous, thus the defective synovial fluid cannot perform its role in cushioning the knee joint, leading to cartilage wearing down with time – page 3). Moreover, please provide full name for EE, PS, and ZP when it appears the first time in the manuscript. Finally, the conclusion is poorly supported by the results.

Author Response

Comments and Suggestions for Authors

Although the work is interesting and well structured, it is necessary to correct the English language because many sentences are not clear (e.g. Although these drugs are recommended by the societies of rheumatology societies and government agencies as they are considered as symptom managing rapid-acting drugs in OA and, treatment of the actual cause of pain remains neglected –page 1; HA (also known as hyaluronan) is utilized in joint diseases treatment, such as OA and rheumatoid arthritis [13], where in these disorders, both the concentration and molecular weight of HA in the synovial fluid have declined and the synovial fluid become less elastic and less viscous, thus the defective synovial fluid cannot perform its role in cushioning the knee joint, leading to cartilage wearing down with time – page 3). Moreover, please provide full name for EE, PS, and ZP when it appears the first time in the manuscript. Finally, the conclusion is poorly supported by the results.

Response: The reviewer’s comments have been addressed. The sentences have been revised and rephrased as requested. The conclusion section was re-written with the results mentioned.

Round 2

Reviewer 1 Report

The authors correctly reviewed the work and answered all questions comprehensively.